# Comparison of Biological Features of Wild European Rabbit Mesenchymal Stem Cells Derived from Different Tissues

**DOI:** 10.3390/ijms23126420

**Published:** 2022-06-08

**Authors:** Alexandra Calle, María Zamora-Ceballos, Juan Bárcena, Esther Blanco, Miguel Ángel Ramírez

**Affiliations:** 1Departamento de Reproducción Animal, Instituto Nacional de Investigación y Tecnología Agraria y Alimentaria (INIA), CSIC, 28040 Madrid, Spain; calle.alexandra@inia.csic.es; 2Centro de Investigación en Sanidad Animal (CISA), Instituto Nacional de Investigación y Tecnología Agraria y Alimentaria (INIA), CSIC, 28130 Madrid, Spain; zamora.maria@inia.csic.es (M.Z.-C.); barcena@inia.csic.es (J.B.); blanco@inia.csic.es (E.B.)

**Keywords:** endangered species, European rabbit, mesenchymal stem cells

## Abstract

Although the European rabbit is an “endangered” species and a notorious biological model, the analysis and comparative characterization of new tissue sources of rabbit mesenchymal stem cells (rMSCs) have not been well addressed. Here, we report for the first time the isolation and characterization of rMSCs derived from an animal belonging to a natural rabbit population within the native region of the species. New rMSC lines were isolated from different tissues: oral mucosa (rOM-MSC), dermal skin (rDS-MSC), subcutaneous adipose tissue (rSCA-MSC), ovarian adipose tissue (rOA-MSC), oviduct (rO-MSC), and mammary gland (rMG-MSC). The six rMSC lines showed plastic adhesion with fibroblast-like morphology and were all shown to be positive for CD44 and CD29 expression (characteristic markers of MSCs), and negative for CD34 or CD45 expression. In terms of pluripotency features, all rMSC lines expressed NANOG, OCT4, and SOX2. Furthermore, all rMSC lines cultured under osteogenic, chondrogenic, and adipogenic conditions showed differentiation capacity. In conclusion, this study describes the isolation and characterization of new rabbit cell lines from different tissue origins, with a clear mesenchymal pattern. We show that rMSC do not exhibit differences in terms of morphological features, expression of the cell surface, and intracellular markers of pluripotency and in vitro differentiation capacities, attributable to their tissue of origin.

## 1. Introduction

The European rabbit (*Oryctolagus cuniculus*) originated in the Iberian Peninsula. The species expanded through anthropogenic dispersal to Europe, Australia, New Zealand, the Americas, and North Africa [1]. A single origin of domestication led to lower levels of genetic diversity within domestic rabbits than those found in wild populations [2]. Thus, natural populations within the European rabbit native region, the Iberian Peninsula, are considered a reservoir of genetic diversity [3]. Over the last 70 years, the European rabbit populations, both in its native and non-native ranges, have suffered declining population trends as reflected in the successive declaration statuses by IUCN: 1996, lower risk/least concern (LR/LC); 2008, near threatened (NT); and 15 August 2018, endangered (https://www.iucnredlist.org/species/41291/170619657 (accessed on 15 August 2018)). The emergence of two viral diseases, myxomatosis and rabbit hemorrhagic disease (RHD), was the primary cause of the decline [4,5,6,7]. In this context, the European rabbit provides a suitable system to study host–viral pathogen interactions, coevolution, and ecological effects [8,9,10,11].

The European rabbit has been extensively studied as a laboratory model for human diseases in both biomedical and fundamental research [12]. After mice (60.9%) and rats (13.9%), the laboratory rabbit is the third most used mammal model in animal experimentation (3.12%) within the EU. Furthermore, the highest increase in comparison to the previous report in 2008 is noted for fish and rabbits (EU report 05 December 2013: https://eur-lex.europa.eu/legal-content/EN/TXT/PDF/?uri=CELEX:52013DC0859 (accessed on 5 December 2013)). However, our current knowledge of important aspects, such as rabbit immunogenetics or the characterization of mesenchymal stem cells (MSCs), lags behind that of other relevant model species such as the mouse.

Mesenchymal stem cells, also known as multipotent stromal cells, are multipotent cells with great therapeutic value due to their usefulness in cell treatment for regenerative medicine and tissue engineering [13]. Because of their immunomodulatory properties, MSCs are widely used in therapy [14]. However, their application in infectious viral diseases has received less attention. In vitro and in vivo, MSCs are susceptible to infection by a wide range of RNA and DNA viruses [15,16,17,18]. Despite their predisposition for the entry of some viruses, there is evidence that MSCs can reduce viral infection by upregulating antiviral mechanisms. Hepatitis C virus replication has been shown to be inhibited by MSC-released miRNAs [19]. MSCs have the ability to interact with and influence both innate and adaptive immune cells, potentially altering the outcome of a viral infection response. While there is information in the literature about the use of MSCs and/or their secretome as a therapy in a variety of animal models of viral diseases, their use in rabbits has yet to be evaluated [20].

MSCs with stem cell characteristics such as plastic adherence, multilineage differentiation capacity, MSC marker expression, and pluripotent gene expression have been found in some rabbit tissues (adipose, gingival, bone marrow, Wharton’s jelly, amniotic fluid) [21,22,23,24]. MSCs can be found in postnatal organs and tissues, but each source has a different degree of differentiation potential and expression of a different set of stem cell-related markers, as well as other important characteristics, e.g., high proliferation, immunomodulation, and allo- and xeno-transplantation ability.

Rabbit MSCs (rMSC) have been widely used as preclinical models for orthopedic problems, specifically bone, articular cartilage, cartilage cell therapy, ligament reconstruction, and spinal fusion, as well as for cardiovascular regenerative medicine strategies [25,26,27,28,29]. Although rMSCs are being investigated in these models, they have not been fully characterized in terms of immunophenotype and differentiation potential, and many tissues have not yet been explored as a source of MSC in the rabbit. Moreover, all the previous rMSC reported in the literature have been obtained from domestic New Zealand White (NZW) line rabbits. Here, we report for the first time the isolation and characterization of rMSC obtained from a female specimen belonging to a wild rabbit population of the species native range, the Iberian Peninsula.

The main goal of the present work was to perform the characterization of rMSC lines isolated from different tissues in terms of morphological features, expression of mesenchymal and pluripotency-related markers, and in vitro chondrogenic, osteogenic, and adipogenic differentiation capacities.

## 2. Results

### 2.1. Morphological Features of Different rMSC

Figure 1 shows that we were able to successfully isolate rMSC from an adult female wild rabbit’s ovarian adipose tissue, subcutaneous adipose tissue, dermal skin, oral mucosa, oviduct, and mammary gland.

rMSCs from all six sources adhered to the plastic surface of culture dishes in primary culture, displaying a mix of round, spindle, and elongated shape morphologies (Figure 1, upper panels). Cells formed a homogeneous population of fibroblast-like adherent cells after the first cell passage (Figure 1, lower panels).

### 2.2. Immunophenotypic Characterization by Flow Cytometry

For further characterization of all six types of rMSCs, some characteristic cell surface markers were assessed by flow cytometry (Figure 2). Cell surface CD44, a characteristic marker of rMSCs, was expressed at detectable levels in all cell lines. In none of the rMSC lines was the expression of hematopoietic lineage markers, such as CD34 and CD45, detected.

### 2.3. Gene Expression of Surface and Pluripotency Markers Detected by RT-PCR

RT-PCR analyses confirmed the expression in all rMSCs lines of those CD surface markers that are characteristic of MSCs (CD29 and CD44) (Figure 3). In addition, CD34 and CD45, hematopoietic lineage markers, could not be detected at the mRNA level in the tested samples. Pluripotent markers OCT4, SOX2, and NANOG were detected by RT-PCR to be expressed in all rMSC lines.

### 2.4. In Vitro Differentiation of rMSCs

As shown in Figure 4, all rMSC lines cultured under chondrogenic conditions showed the presence of acidic proteoglycan synthesized by chondrocytes that were demonstrated at the monolayer of cells by Alcian blue staining, which appeared in most cases as stained nodules typical from cartilaginous tissue. Cells cultured under osteogenic conditions presented remarkable calcium deposits, indicating a high osteogenic differentiation potential of these lines. All rMSC lines obtained were also able to differentiate to adipocytes when cultured under adipogenic conditions, presenting the formation of cytoplasmic lipid droplets visualized after the Oil red O solution stain.

## 3. Discussion

The International Society for Cellular Therapy proposed three criteria to define the minimal characteristics of MSCs in 2006 with the goal of standardization [30]: When maintained in standard culture conditions using tissue culture flasks, they should exhibit plastic adherence; more than 95% of the MSC population should express specific markers, such as CD105, CD73, and CD90, and be negative for CD45, CD34, CD14 or CD11b, CD79 or CD19, and HLA class II; and they should be able to differentiate to osteoblasts, adipocytes, or chondroblasts in vitro under standard differentiating conditions as demonstrated.

Our findings revealed that rabbit MSCs could be isolated and expanded in vitro from visceral adipose depots, subcutaneous fat, dermal skin, oral mucosa, oviduct, and mammary gland tissues of an adult female wild rabbit. Our results also demonstrate that rOA-MSCs, rSCA-MSCs, rDS-MSCs, rOM-MSC, rO-MSC, and rMG-MSCs shared similar characteristics in terms of morphology, expression of mesenchymal and pluripotency-related markers, and differentiation ability into chondrocytes, osteocytes, and adipocytes. Passaged cells had a more homogeneous morphology and formed colonies as the culture progressed, compared to primary cultures. As we have shown in previous studies on porcine and bovine species, these morphological observations indicate that the isolated cells may contain both mature and progenitor populations [31,32,33].

Regarding cell surface markers characteristic of rMSC, contradictory results have been described in rabbits, mainly related to CD105, CD73, and CD90, as summarized by Zomer et al. [34]. Martínez-Lorenzo et al. found CD73, CD90, and CD105 expression percentages of 1.6, 40.1, and 20.5 in rabbit MSC, respectively [35]. However, Lee et al. later reported that CD73, CD90, and CD105 were present on human MSCs but not on rabbit MSCs [36]. Kovac et al. found a low positive for CD90 but a negative for CD73 and CD105 [24]. Considering the above, a panel of antibodies that have shown high sensitivity and repeatability and confirmed these results through RT-PCR assessment of expression at the mRNA level was selected [34]. Our data demonstrate that all rMSCs isolated were negative for CD34 and CD45 but positive for CD44, CD29, NANOG, OCT4, and SOX2, coinciding with the results reported in the literature for rMSCs isolated from different tissues: adipose [21], gingival [22], amniotic fluid [24], and bone marrow [26,37]. OCT4 and SOX2 are both naturally expressed in embryonic and adult stem cells. They are, however, typically expressed at low levels in early-passage MSCs and gradually decrease as the number of passages increases [38,39]. Han et al. co-expressed OCT4 and SOX2 in MSC to confer to these cells enhanced proliferation and differentiation capabilities [40]. Furthermore, OCT4 and NANOG are required for preserving MSC characteristics as well as maintaining pluripotency in embryonic stem cells [41]. Specifically, the expression of pluripotency markers in MSC has already been reported in rabbit [21].

The basic in vitro trilineage differentiation capacity of rMSC, that is adipocytes, osteocytes, and chondrocytes has been reported in the literature only for adipose [34] and for bone marrow rMSC [26,27,36]. Our results show that all isolated rMSC lines, regardless of their tissue of origin, were able to differentiate into the three mesodermal lineages, with lipid droplet accumulation, calcium deposits, or the presence of proteoglycan in response to in vitro adipogenic, osteogenic, or chondrogenic stimuli, respectively.

The gene expression profile of adipose MSC from subcutaneous fat in Zucker rats was found to be distinct from the gene expression profiles of the other four visceral designated depots (epicardial, epididymal, mesenteric, and retroperitoneal), which clustered together but were not identical [42]. Moreover, differences between adipose-MSC from mesenteric and omental depots have been documented [43]. These findings show that individual visceral “subdepots” should not be considered interchangeable. Moreover, increased visceral (central) fat is more pathological, resulting in insulin resistance and chronic inflammation, whereas subcutaneous (peripheral) fat is more beneficial in systemic metabolism [44]. These differences are cell-intrinsic and persist in culture [43,45], revealing the mechanisms underlying the pathophysiological differences between adipose depots. Individual adipose depots, which function as separate “mini-organs,” are formed and maintained by distinct A-MSC populations, with consistent differences in adipogenic potential and function later in life [46,47]. We have however not observed major differences between rMSCs isolated from subcutaneous adipose tissue or ovarian adipose depot, although it would be interesting to perform a more in-depth characterization.

Rabbit embryo and fetoplacental development are similar to human development, making the rabbit an excellent model for studying embryo–maternal communication, the effects of maternal metabolic disorders on offspring development and long-term health [48], and the development of in vitro models of pathogen infection during pregnancy. In 2009, Jazedje et al. showed for the first time that human fallopian tubes are a rich additional source of MSCs, and these cells were designated as human tube MSCs (htMSCs) [49]. The presence of mesenchymal cell populations in this part of the female reproductive system has only been documented in humans until now [50,51,52,53]. Our rO-MSC will allow the development of both physiological and pathological in vitro models.

Due to the remarkable cyclical changes in proliferation, lactation, and involution that occur in the breast tissue throughout life and pregnancy, the field of mammary gland physiology has been historically interested in stem cell biology. Shackleton et al. demonstrated that a single cell lacking hematopoietic and endothelial antigens and expressing high levels of CD29 could generate an entire mammary gland [54]. Following this description, in rMG-MSC, we found no expression of the hematopoietic lineage markers CD34 and CD45, while they expressed CD29 and CD45 as well as pluripotency markers (NANOG, OCT4, and SOX2). Mammary gland stem cells have subsequently been shown to play key roles in both regenerations of the mammary gland and the development of mammary gland tumors. CD44 has been used as a marker of cancer-initiating cells in various cancers, including prostate, pancreas, and colon [55,56]. Considering the above, rMG-MSC is promising in vitro model for the study of both breast physiology and breast cancer, since it expressed CD44. Until now, the presence of MSCs in the mammary gland has only been studied in mice [57,58,59,60].

The comparison of biological features of wild European rabbit mesenchymal stem cells derived from different tissues shows that rMSC does not exhibit differences in terms of morphological features, expression of the cell surface, and intracellular markers of pluripotency and in vitro differentiation capacities, attributable to their tissue of origin.

The new tissue sources of rabbit mesenchymal stem cells used in this study will allow significant advances in already widely used preclinical models and will open the door to new models as mentioned above. Furthermore, the isolation of new rMSC lines from target tissues for the species’ main viral pathogens (myxoma virus and rabbit hemorrhagic disease virus (RHDV)) will allow for an in-depth analysis of their susceptibility to viral infection as well as their possible role in the regulation of host antiviral response, avoiding the use of an animal model. The inability of RHDV to be propagated in vitro has slowed research into its pathogenesis, translation, and replication mechanisms, and rMSC, as with other viruses, could be a solution to this problem.

## 4. Materials and Methods

### 4.1. Isolation and Culture of rMSCs

Different tissue samples were obtained postmortem from an adult female wild rabbit (*Oryctolagus cuniculus*).

For the isolation of primary mesenchymal cultures from oral mucosa (rOM-MSC), dermal skin tissue (rDS-MSC), subcutaneous adipose tissue (rSCA-MSC), ovarian adipose tissue (rOA-MSC), oviduct (rO-MSC), and mammary gland (rMG-MSC), the procedures are detailed in Calle et al. [31]. Briefly, the collected samples were minced before the tissues were incubated in a collagenase type I solution. Thereafter, a volume of culture medium was added to block the action of collagenase. The resulting pellets were resuspended in a culture medium, plated in a 100-mm^2^ tissue culture dish (Jet-Biofil, Guangzhou, China), and incubated in an atmosphere of humidified air and 5% CO_2_ at 37 °C. The culture medium was changed every 48–72 h. Isolated colonies of putative rMSCs were apparent after 5–6 days in culture and were maintained in a growth medium until ~75% confluence. The cells were then treated with 0.05% trypsin–EDTA (T/E) and further cultured for subsequent passage in 100-mm^2^ dishes at 50,000 cells/cm^2^.

### 4.2. Immunocytochemical Analysis by Flow Cytometry

Cell cultures at 80–90% confluence were detached and fixed with 4% paraformaldehyde for 10 min. Stainings were performed as detailed in Calle et al. [31], employing anti-CD44 (clone W4/86, Bio-Rad-MCA806GA. Hercules, CA, USA); anti-CD34 (Mouse monoclonal, Invitrogen-MA1-10202. Rockford, IL, USA), and anti-CD45 (clone L12/201, Bio-Rad-MCA808GA, Hercules, CA, USA). Appropriated secondary staining was performed using F(ab’)2-Goat anti-Mouse IgG (H + L) Cross-Adsorbed Secondary Antibody, APC (A10539) from Life Technologies (Carlsbad, CA, USA). Negative control samples corresponded to samples in which the primary antibody was omitted. Samples were analyzed in a FACSCalibur (BD Biosciences, San Jose, CA, USA) using Flow-JoX Software^®^ version 10.0.7r2 (TreeStar, Ashland, Oregon, USA).

### 4.3. RT-PCR Analysis of Characteristic Mesenchymal mRNA Expression in rMSC

Total RNA was extracted from 2.5 × 10^6^ cells by RNeasy Mini kit (Qiagen, Dusseldorf, Germany), according to the manufacturer’s instructions. The RNA starting template was analyzed for integrity and quantity by Nanodrop test, and 600 ng were used in the RT-PCR reaction. cDNA was synthesized and amplified using OneStep RT-PCR Kit (Qiagen), according to the manufacturer’s instructions and using the following gene-specific primers: Rabbit CD29, forward (AGAATGTCACCAACCGTAGCA), reverse (CACAAAGGAGCCAAACCCA); rabbit CD44, forward (TCATCCTGG-CATCCCTCTTG), reverse (CCGTTGCCATTGTTGATCAC); rabbit CD45, forward (TACTCTGCCTCCCGTTG), reverse (GCTGAGTGTCTGCGTGTC); rabbit CD34, forward (TTTCCTCATGAACCGTCGCA), reverse (CGTGTTGTCTTGCGGAATGG); NANOG, forward (GCCAGTCGTGGAGTAACCAT), reverse (CTGCATGGAGGACTGTAGCA); OCT4, forward (GAGGAGTCCCAGGACATGAA), reverse (GTGGTTTGGCTGAACACCTT); SOX2, forward (CAGCTCGCAGACCTACATGA), reverse (TGGAGTGGGAGGAAGAGGTA) [24]. PCR products were separated on a 1.5% agarose gel (MB Agarose, Bio-Rad, Hercules, CA, USA) stained with GelRed^®^ Nucleic Acid Gel Stain (Biotium, #41002).

### 4.4. In Vitro Differentiation Potential Assay

For adipogenic, osteogenic, and chondrogenic differentiation, cells were cultured according to the manufacturer’s instructions of the StemPro^®^ adipogenesis, osteogenesis, or chondrogenesis differentiation kits, respectively (Thermo Fisher Scientific, Meridian Road, Rockford, IL, USA) and analyzed as detailed in Calle et al. [31].

## 5. Conclusions

In summary, this study describes the isolation and characterization for the first time of cell lines from different tissue origins with a clear mesenchymal pattern, derived from a natural rabbit population within the native region of the species. We show that rMSCs do not present differences in terms of morphological features, expression of the cell surface, and intracellular markers of pluripotency and in vitro chondrogenic, osteogenic, and adipogenic differentiation capacities, attributable to their tissue of origin.

## Figures and Tables

**Figure 1 ijms-23-06420-f001:**
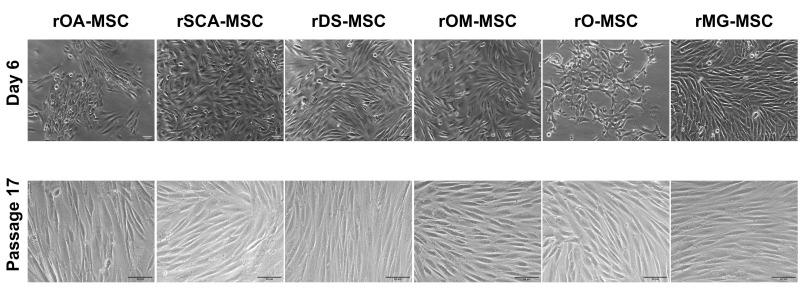
Phase-contrast images of different rabbit rMSC lines at 6 days of culture of Passage 0 (upper panels; ×100 magnification) and Passage 17 (lower panels; ×200 magnification). Bars = 50 μm.

**Figure 2 ijms-23-06420-f002:**
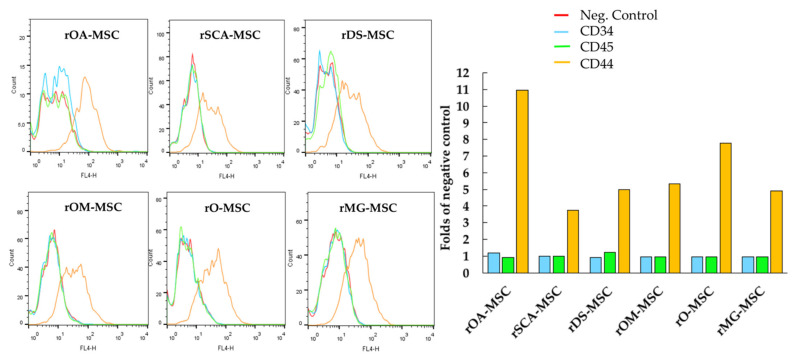
Analysis by flow cytometry of the expression levels of cell surface markers CD34, CD44, and CD45 in rOA-MSC, rSCA-MSC, rDS-MSC, rOM-MSC, rO-MSC and rMG-MSC. Plots depict the log of fluorescence intensity for each sample. Bar charts correspond to the mean fluorescence intensity (folds of negative control in the absence of primary antibody) for each sample.

**Figure 3 ijms-23-06420-f003:**
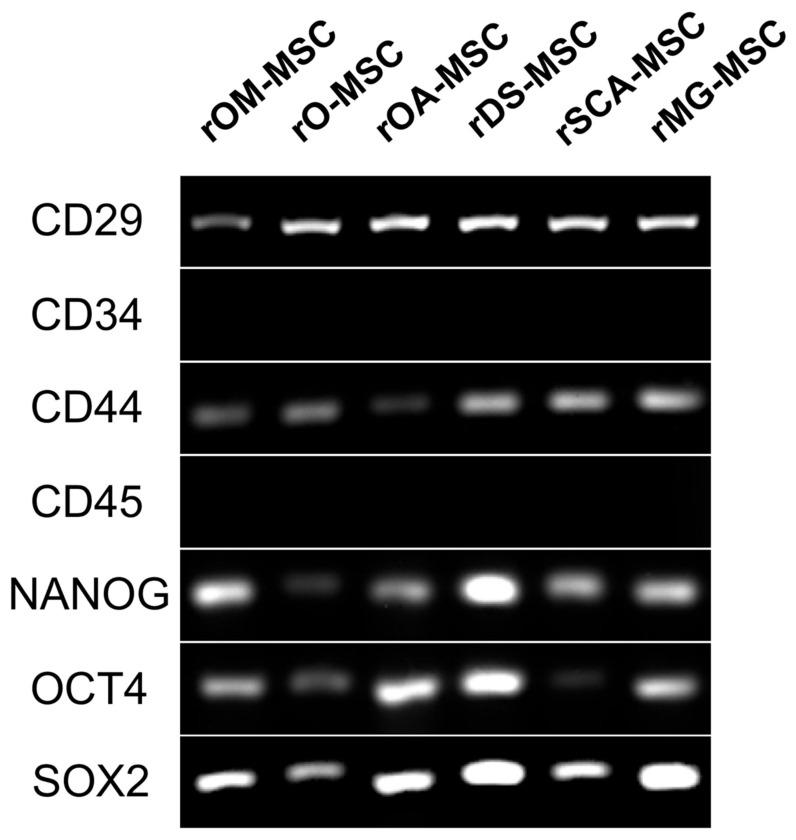
RT-PCR analysis of surface and pluripotency marker gene expression in the different rMSC. A panel of surface markers including CD29, CD34, CD44, and CD45 was used for the identification of rMSC. Pluripotency genes expression (NANOG, OCT4, and SOX2) was detected in the different rMSC.

**Figure 4 ijms-23-06420-f004:**
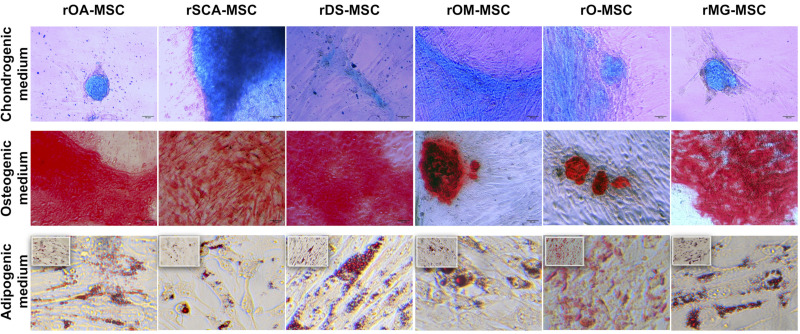
In vitro differentiation of rMSC to different lineages. Images show Alcian blue staining of acidic proteoglycan in cells cultured in chondrogenic differentiation medium (**top** panel); Alizarin Red S staining of calcium deposits in cells cultured in osteogenic differentiation medium (**middle** panels); and Oil red O staining of lipid droplets in cells cultured in adipogenic differentiation medium (**bottom** panels). Bright-field images were acquired with 200× magnification (bars = 50 μm) for top and middle panels and with 400× magnification (bars = 100 μm) for bottom panels.

## Data Availability

The data that support the findings of this study are available from the corresponding author upon reasonable request.

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
