# Peer review of "Comparison of Biological Features of Wild European Rabbit Mesenchymal Stem Cells Derived from Different Tissues"

_ijms, 2022, doi:10.3390/ijms23126420_

Round 1
Reviewer 1 Report
In the article: “Comparison of biological features of wild European rabbit mesenchymal stem cells derived from different tissues” the authors explained the potential application of mesenchymal cells isolated from different tissue origins.
The authors clearly explain the rational of the study, the performed experiments and discussed the obtained results. However, we would like to invite the authors to clarify some minor points:
- The authors should better explain how these animal derived cells can be considered in a scientific world where the animals are being used less for experimentation;
- The authors should better explain the importance of mesenchymal cells and their potential application into regenerative medicine thus the ability to differentiate in several cell lines. In this respect the citation of the following scientific manuscript should be useful: Alessio N, Stellavato A, Aprile D, Cimini D, Vassallo V, Di Bernardo G, Galderisi U, Schiraldi C. Timely Supplementation of Hydrogels Containing Sulfated or Unsulfated Chondroitin and Hyaluronic Acid Affects Mesenchymal Stromal Cells Commitment Toward Chondrogenic Differentiation. Front Cell Dev Biol. 2021 Apr 12;9:641529. doi: 10.3389/fcell.2021.641529;
- Please check the check punctuation and spaces;
- The authors performed the RT-PCR in order to analyze the expression of specific biomarkers of mesenchymal cells, why they did not also consider the housekeeping?
- In the Figure 2, there are not standard deviation in the graphs, please specify how many analyses were run.
Author Response
Reviewer 1:
In the article: “Comparison of biological features of wild European rabbit mesenchymal stem cells derived from different tissues” the authors explained the potential application of mesenchymal cells isolated from different tissue origins.
The authors clearly explain the rational of the study, the performed experiments and discussed the obtained results. However, we would like to invite the authors to clarify some minor points:
- The authors should better explain how these animal derived cells can be considered in a scientific world where the animals are being used less for experimentation;
Indeed, we agree with the reviewer, and avoiding the use of animals in research is one of the spearheads shared by all our lines of research. The isolation of new rabbit MSC lines from target tissues for the species' main viral pathogens (myxoma virus and rabbit hemorrhagic disease virus (RHDV)) will allow an in-depth analysis of their susceptibility to viral infection as well as their possible role in the regulation of host antiviral response, avoiding the use of an animal model. The inability of RHDV to be propagated in vitro has slowed research into its pathogenesis and replication mechanisms, and rabbit MSC, as with other viruses, could be a solution to this problem. In addition, the new tissue sources of rabbit mesenchymal stem cells characterized in this study will allow significant advances in already widely used preclinical models and open the door to new models for the study of both breast physiology and breast cancer.
Following the reviewer's suggestion, a paragraph has been rewritten in the discussion section (line 221).
“The new tissue sources of rabbit mesenchymal stem cells used in this study will allow significant advances in already widely used preclinical models and will open the door to new models as mentioned above. Furthermore, the isolation of new rMSC lines from target tissues for the species' main viral pathogens (myxoma virus and rabbit hemorrhagic disease virus (RHDV)) will allow for an in-depth analysis of their susceptibility to viral infection as well as their possible role in the regulation of host antiviral response, avoiding the use of an animal model. The inability of RHDV to be propagated in vitro has slowed research into its pathogenesis, translation, and replication mechanisms, and rMSC, as with other viruses, could be a solution to this problem”.
- The authors should better explain the importance of mesenchymal cells and their potential application into regenerative medicine thus the ability to differentiate in several cell lines. In this respect the citation of the following scientific manuscript should be useful: Alessio N, Stellavato A, Aprile D, Cimini D, Vassallo V, Di Bernardo G, Galderisi U, Schiraldi C. Timely Supplementation of Hydrogels Containing Sulfated or Unsulfated Chondroitin and Hyaluronic Acid Affects Mesenchymal Stromal Cells Commitment Toward Chondrogenic Differentiation. Front Cell Dev Biol. 2021 Apr 12; 9:641529. doi: 10.3389/fcell.2021.641529;
Following the reviewer's suggestion, a paragraph was added and another was rewritten in the introduction section. (lines 49 and 69). In addition, the bibliography proposed by the reviewer has been included.
“Mesenchymal stem cells, also known as multipotent stromal cells, are multipotent cells with great therapeutic value due to their usefulness in cell treatment for regenerative medicine and tissue engineering [13]”.
“Rabbit MSCs (rMSC) have been widely used as preclinical models for orthopedic problems, specifically bone, articular cartilage, cartilage cell therapy, ligament reconstruction, and spinal fusion, as well as for cardiovascular regenerative medicine strategies [25–29]”.
- Please check the check punctuation and spaces;
The entire document has been verified for grammatical and linguistic errors.
- The authors performed the RT-PCR in order to analyze the expression of specific biomarkers of mesenchymal cells, why they did not also consider the housekeeping?
The housekeeping gene method of normalization was not used since it should have required a previous selection and validation of the constant expression levels of the possible reference genes between cells of different tissues to avoid misinterpretations of the results. Since the aim of our analyses was just to check the expression or not of different biomarkers, we performed conventional one-step RT-PCR using an approach based on the normalization of total RNA (quality and quantity were measured before the RT-PCR). The RT-PCR results are consistent with the FACS data, further corroborating the appropriateness of our approach.
- In the Figure 2, there are not standard deviation in the graphs, please specify how many analyses were run.
Bar charts correspond to the mean fluorescence intensity (folds of negative control in the absence of primary antibody) for each sample and therefore there is no error bar. The selected antibodies panel has shown high sensitivity and repeatability and we have also confirmed these results through RT-PCR assessment of expression at the mRNA level. Our flow cytometry data are in agreement with RT-PCR and with previous reports in the literature for rMSCs isolated from different tissues: adipose, gingival, amniotic fluid, and bone marrow. They demonstrate that all rMSCs were negative for CD34 and CD45, but positive for CD44, CD29, NANOG, OCT4, and SOX2.
Following the reviewer's suggestion, the caption of Figure 2 has been rewritten (line 102).
“Figure 2. Analysis by flow cytometry of the expression levels of cell surface markers CD34, CD44, and CD45 in rOA-MSC, rSCA-MSC, rDS-MSC, rOM-MSC, rO-MSC, and rMG-MSC. Plots depict the log of fluorescence intensity for each sample. Bar charts correspond to the mean fluorescence intensity (folds of negative control in the absence of primary antibody) for each sample.”

Reviewer 2 Report
This study aims to compare the biological features the MSCs from different origins from European rabbit. In this study, the authors isolated MSCs from different origins of European rabbit oral mucosa (rOM-MSC), dermal skin (rDS-MSC), subcutaneous adipose tissue (rSCA-MSC), ovarian adipose tissue (rOA-MSC), oviduct (rO-MSC), and mammary gland (rMGMSC). The trilineage differentiations including chondrogenesis, osteogenesis, and adipogenesis were tested. Moreover, the surface antigens were also tested
1. The antibodies used to detect the rabbit origin MSCs are difficult to find. Since the authors can successfully detect the MSC derived from rabbits, the catalog numbers of antibodies used to detect the surface antigens of rabbit MSCs should be provided.
2. MSCs are usually considered as multipotent due to they can differentiate into cells derived from mesoderm. However, the authors detect the pluripotent markers. This should be discussed.
3. Sine the authors are to compare the biological features of MSCs derived from different origins, the differences of these MSCs should be discussed.
4. Have authors performed any experiments about the viral infection susceptibility of HRDV on these MSCs? Or, any in vitro model was developed (something like organoid ) using these 6 different origins of MSCs and differences among these cells should be compared.
Author Response
Reviewer 2:
This study aims to compare the biological features the MSCs from different origins from European rabbit. In this study, the authors isolated MSCs from different origins of European rabbit oral mucosa (rOM-MSC), dermal skin (rDS-MSC), subcutaneous adipose tissue (rSCA-MSC), ovarian adipose tissue (rOA-MSC), oviduct (rO-MSC), and mammary gland (rMGMSC). The trilineage differentiations including chondrogenesis, osteogenesis, and adipogenesis were tested. Moreover, the surface antigens were also tested
- The antibodies used to detect the rabbit origin MSCs are difficult to find. Since the authors can successfully detect the MSC derived from rabbits, the catalog numbers of antibodies used to detect the surface antigens of rabbit MSCs should be provided.
The specific reference of each antibody used has now been included, as suggested by the reviewer (Line 248).
“Stainings were performed as detailed in Calle et al. [30], employing anti‐CD44 (clone W4/86, Bio‐Rad-MCA806GA); anti‐CD34 (Mouse monoclonal, Invitrogen-MA1-10202), and anti-CD45 (clone L12/201, Bio‐Rad-MCA808GA)”.
- MSCs are usually considered as multipotent due to they can differentiate into cells derived from mesoderm. However, the authors detect the pluripotent markers. This should be discussed.
Following the reviewer's suggestion, a paragraph has been included in the discussion section (line 165).
“OCT4 and SOX2 are both naturally expressed in embryonic and adult stem cells. They are, however, typically expressed at low levels in early-passage MSCs and gradually decrease as the number of passages increases [38, 39]. Han et al co-expressed OCT4 and SOX2 in MSC to confer to these cells enhanced proliferation and differentiation capabilities [40]. Furthermore, OCT4 and NANOG are required for preserving MSC characteristics as well as maintaining pluripotency in embryonic stem cells [41]. Specifically, the expression of pluripotency markers in MSC has already been reported in rabbit [21]”.
- Sine the authors are to compare the biological features of MSCs derived from different origins, the differences of these MSCs should be discussed.
Following the reviewer’s suggestion, a paragraph has been included in the discussion section (line 217).
“The comparison of biological features of wild European rabbit mesenchymal stem cells derived from different tissues shows that rMSC does not exhibit differences in terms of morphological features, expression of the cell surface, and intracellular markers of pluripotency and in vitro differentiation capacities, attributable to their tissue of origin”.
- Have authors performed any experiments about the viral infection susceptibility of HRDV on these MSCs? Or, any in vitro model was developed (something like organoid) using these 6 different origins of MSCs and differences among these cells should be compared.
We appreciate the reviewer’s interest in determining the susceptibility of different rMSCs to RHDV infection. We are currently conducting these analyses and hope to present the results to the scientific community as soon as possible.

Reviewer 3 Report
Since the Authors wrote that they reported for the first time the isolation and characterization of rMSCs derived from an animal belonging to a natural rabbit population, the proper MSC markers estimation must be done.
Please look at
Minimal criteria for defining multipotent mesenchymal stromal cells. The International Society for Cellular Therapy position statement
Author Response
Reviewer 3:
Since the Authors wrote that they reported for the first time the isolation and characterization of rMSCs derived from an animal belonging to a natural rabbit population, the proper MSC markers estimation must be done.
Please look at
Minimal criteria for defining multipotent mesenchymal stromal cells. The International Society for Cellular Therapy position statement
M Dominici 1, K Le Blanc, I Mueller, I Slaper-Cortenbach, Fc Marini, Ds Krause, Rj Deans, A Keating, Dj Prockop, Em Horwitz MSC must express CD105, CD73 and CD90, and lack expression of CD45, CD34, CD14 or CD11b, CD79alpha or CD19 and HLA-DR surface molecules. Markers you did – positive for CD44 and CD29 expression (characteristic markers of MSCs), and negative for CD34 or CD45 are not sufficient.
We greatly appreciate the reviewer’s input. Indeed, Dominici et al is an especially relevant reference for the correct characterization of MSC, so we now include a paragraph about it in the discussion chapter (Line 135).
“The International Society for Cellular Therapy proposed three criteria to define the minimal characteristics of MSCs in 2006 with the goal of standardization [30]: When maintained in standard culture conditions using tissue culture flasks, they should exhibit plastic adherence; more than 95 percent of the MSC population should express specific markers such as CD105, CD73, and CD90, and be negative for CD45, CD34, CD14 or CD11b, CD79 or CD19, and HLA class II; and they should be able to differentiate to osteoblasts, adipocytes, or chondroblasts in vitro under standard differentiating conditions as demonstrated”.
In the discussion section, a paragraph has been rewritten to clarify that antibodies with high sensitivity and repeatability in the literature were chosen and those data were further confirmed by RT-PCR assessment of expression at the mRNA level (Line 153).
“Regarding cell surface markers characteristic of rMSC, contradictory results have been described in rabbits, mainly related to CD105, CD73, and CD90, as summarized by Zomer et al. [34]. Martínez-Lorenzo et al found CD73, CD90, and CD105 expression percentages of 1.6, 40.1, and 20.5 in rabbit MSC, respectively [35]. However, Lee et al. later reported that CD73, CD90, and CD105 were present on human MSCs but not on rabbit MSCs [36]. Kovac et al. found a low positive for CD90 but a negative for CD73 and CD105 [24]”.

Round 2
Reviewer 2 Report
All my questions have been answered.
Reviewer 3 Report
none